# Accurate Simulation of Ice and Snow Runoff for the Mountainous Terrain of the Kunlun Mountains, China

**Yongchao Duan** [1,2,3,4,5,6], **Tie Liu** [1,2,3,5,*] , **Fanhao Meng** [7], **Ye Yuan** [1,2,3], **Min Luo** [7], **Yue Huang** [1,2,3], **Wei Xing** [3], **Vincent Nzabarinda** [1,2,3] and **Philippe De Maeyer** [4,5,6]

1   State Key Laboratory of Desert and Oasis Ecology, Xinjiang Institute of Ecology and Geography, Chinese Academy of Sciences, Urumqi 830011, China; duanyongchao13@mails.ucas.ac.cn (Y.D.); yuanye183@mails.ucas.ac.cn (Y.Y.); huangy@ms.xjb.ac.cn (Y.H.); vincentnzabarinda@mails.ucas.ac.cn (V.N.)
2   Key Laboratory of GIS & RS Application Xinjiang Uygur Autonomous Region, Urumqi 830011, China
3   University of Chinese Academy of Sciences, Beijing 100049, China; xingwei@ms.xjb.ac.cn
4   Department of Geography, Ghent University, 9000 Ghent, Belgium; philippe.demaeyer@ugent.be
5   Sino-Belgian Joint Laboratory of Geo-information, Urumqi 830011, China
6   Sino-Belgian Joint Laboratory of Geo-information, 9000 Gent, Belgium
7   Inner Mongolia Normal University, Hohhot 010022, China; mfh320@imnu.edu.cn (F.M.); luomin@imnu.edu.cn (M.L.)
*   Correspondence: liutie@ms.xjb.ac.cn; Tel.: +86-991-788-5378

**Abstract:** While mountain runoff provides great potential for the development and life quality of downstream populations, it also frequently causes seasonal disasters. The accurate modeling of hydrological processes in mountainous areas, as well as the amount of meltwater from ice and snow, is of great significance for the local sustainable development, hydropower regulations, and disaster prevention. In this study, an improved model, the Soil Water Assessment Tool with added ice-melt module (SWATAI) was developed based on the Soil Water Assessment Tool (SWAT), a semi-distributed hydrological model, to simulate ice and snow runoff. A temperature condition used to determine precipitation types has been added in the SWATAI model, along with an elevation threshold and an accumulative daily temperature threshold for ice melt, making it more consistent with the runoff process of ice and snow. As a supplementary reference, the comparison between the normalized difference vegetation index (NDVI) and the quantity of meltwater were conducted to verify the simulation results and assess the impact of meltwater on the ecology. Through these modifications, the accuracy of the daily flow simulation results has been considerably improved, and the contribution rate of ice and snow melt to the river discharge calculated by the model increased by 18.73%. The simulation comparison of the flooding process revealed that the accuracy of the simulated peak flood value by the SWATAI was 77.65% higher than that of the SWAT, and the temporal accuracy was 82.93% higher. The correlation between the meltwater calculated by the SWATAI and the NDVI has also improved significantly. This improved model could simulate the flooding processes with high temporal resolution in alpine regions. The simulation results could provide technical support for economic benefits and reasonable reference for flood prevention.

**Keywords:** SWAT; ice-melt; flood processes; accumulated temperature; NDVI; Tizinafu River Basin (TRB)

## 1. Introduction

As an important part of the world's high-altitude mountains, glaciers and snow provide abundant water resources [1–3]. Most of the world's great rivers collect source water from glaciers or snow-covered

mountains at high altitudes [4]. Along with the more studies of global climate change [5–8], the studies on glaciers and snow cover have been receiving increasing attention [9]. Although these glaciers and snow covers could provide large amounts of water resources [10], they could also cause serious disasters. When considering ways in which to utilize these resources, the fundamental approach is to determine the amount and features of meltwater at a specified location. In the context of global climate change, more experts have combined remote sensing data of glacier snow cover with hydrological processes [4,11–14] to overcome the data lack situation.

With the background of glacier retreatment in recent decades in the mountainous areas of Central Asia and the Xinjiang region of Northwest China [6,15–21], the importance of hydrological modeling is becoming increasingly evident [22]. Various empirical methods and algorithms have commonly been used in the calculation of ice and snow melt, such as the method of glacier mass balance using the relationship between the volume and surface ratio of glaciers [23–26]. Subsequently, numerous studies have also been conducted on the hydrological processes of glacier-dominated watersheds in the context of future climate change [27–33]. Indeed, some semi-distributed models in combination with conceptual models have also been applied in hydrology-related studies of glacier changes. Moreover, the distributed hydrological models based on physics have also been increasingly utilized in relevant glacier studies, such as calculating the amount of ice and snow melt, simulating river runoff, to determine the relationship between glaciers and the environment [22]. The SWAT model [22,34,35], as the outstanding representative of distributed hydrological models, is based on energy balance, physical balance, and water balance, including soil, land use, meteorological, and hydrological management modules [4,36]. Most importantly, as an open-source hydrological model, SWAT provides a window into model optimization and modification.

Since emphasis has been placed on glacier changes in the context of climate change on a large scale and over a long time series [37–43], some researchers proposed an improved SWAT-based computational framework of critical source areas identification at the lake basin scale, which improved the ability of supplying a more comprehensive delineation of critical source areas [44]. In the studies of the SWAT model in the process of mountainous runoff [45,46], some scholars have tried to evaluate the impacts of temperature and precipitation changes on runoff and streamflow by trying to add ice melting algorithms [4,47,48]. While the hydrological process of meltwater in a watershed scale at a high temporal resolution has rarely been studied. In high-incidence areas of ice and snow melt, flooding simulation with high temporal resolution is indispensable.

In the past, very few studies have been able to perform the hourly scale processes of snow melting and ice melting floods in the alpine region under complex climate conditions. In order to overcome these issues, the topography and climate characteristics of alpine area are fully considered in the discrimination of precipitation form, and the influence of accumulated temperature on precipitation form was added into the new modular. The main goal of this study was to embed a new ice-melt module into the original SWAT model; the segmentation of the ice and snow melt calculation was performed by setting the elevation threshold. Hourly simulation was performed based on the daily time scale in order to delineate the detailed flood process. In order to better explore the relationship between meltwater and local ecology, a comparison between the NDVI of long time series and the calculation of the model's meltwater was conducted. This study can provide technical support and act as an important reference for flood disaster prevention, water conservancy construction, ecological restoration, and other environmental concerns.

## 2. Materials and Methods

### 2.1. Study Area

The study area is the Tizinafu River Basin (TRB), located in the Kunlun Mountains of the southwest portion of the Xinjiang region in Northwest China (Figure 1). The landscape in this region is composed primarily of rock, sand, and gravel, with sparse vegetation. Only a small amount of vegetation exists

in the valleys. The lowest elevation within the basin is 1,400 m, located in the lower part of the basin in Yecheng County area, and the highest elevation is 6,320 m, which is also the source of the river, the Keerake Daban and Yanggai Daban. The total area of the TRB is approximately 5,600 km$^2$. Affected by topographic conditions, the diurnal range in the basin is large. The climate of the TRB can be divided into the melting season and the non-melting season. In spring, temperatures start to rise, snow and glaciers gradually begin to melt, and the river channel is recharged. In the beginning of October, temperatures in the mountainous areas drop rapidly and snowfall occurs in high-altitude areas as the TRB gradually enters the non-melting season.

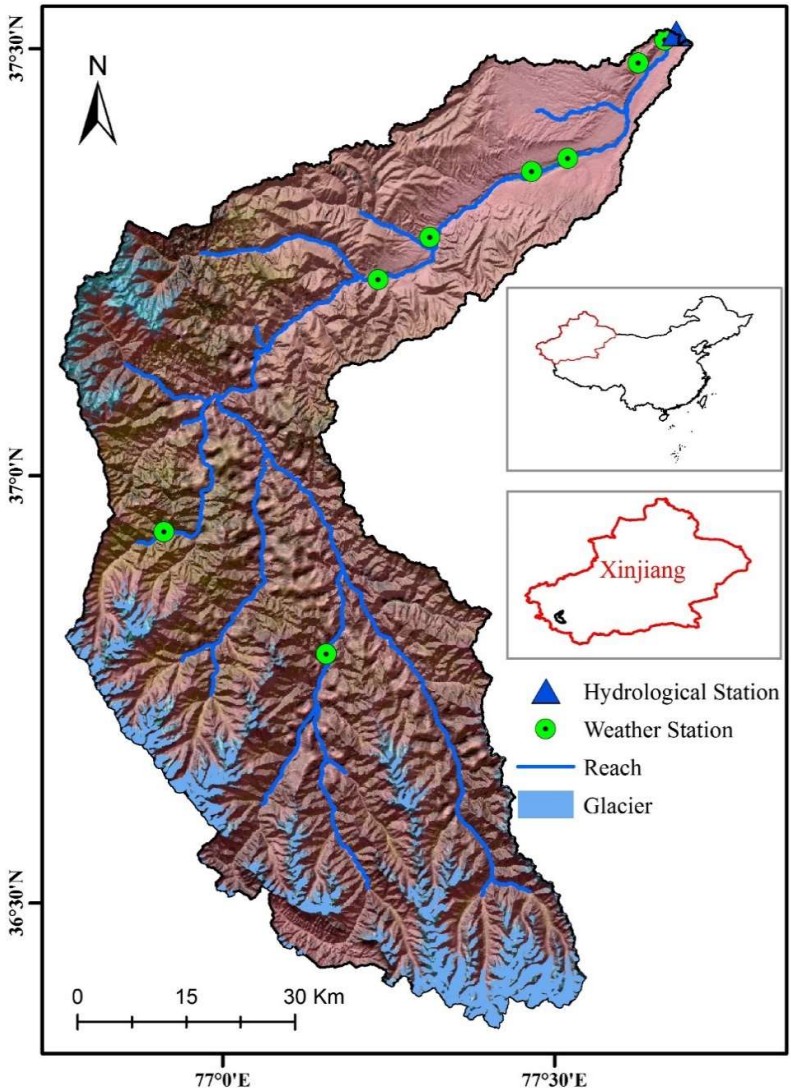

**Figure 1.** Map illustrating the topography and station distribution in the TRB.

*2.2. Materials*

The digital elevation model (DEM), which is obtained from (http://srtm.csi.cgiar.org/) the Shuttle Radar Topography Mission (SRTM), has a spatial resolution of 30 meters and served as the primary data source for the watershed division and model calculation. Both spatial resolutions of the soil type data from the China Soil Category Data Network (https://geodata.pku.edu.cn/) as well as the land use and land cover (LUCC) data from the visual interpretation of imagery (https://www.usgs.gov/products/data-and-tools/real-time-data/remote-land-sensing-and-landsat) were 30 meters. Eight-day snow cover product of the Moderate Resolution Imaging Spectroradiometer (MODIS) is used as

calibration reference and initial conditions (https://modis.gsfc.nasa.gov/data/dataprod/mod10.php). Glacier data from the Randolph Glacier Inventory, are derived from Global Land Ice Measurements from Space (GLIMS) (http://www.glims.org/RGI/). In order to compare the relationship between the calculated meltwater and the NDVI for a long time series, the NDVI data were calculated from 8-day surface reflectance data MOD09Q1 (https://modis.gsfc.nasa.gov/data/dataprod/mod09.php). The ERA-Interim from the European Centre for Medium-Range Weather Forecasts (ECMWF) was used as the meteorological reanalysis data (https://www.ecmwf.int/).

The meteorological and hydrological observation are used as model input based on the national stations. According to the data from 2010 to 2014, the average daily temperature of the river basin was approximately 7 °C, and the average annual discharge was approximately 35.22 $m^3 \cdot s^{-1}$.

The classification and statistics of the land use and land cover (Table 1) shows that the glacier and snowfield take considerable portion of study area. The area of glacier and snowfield in the study region was as high as 469.96 $km^2$, accounting for 8.35% of the total area. Snow and glacial meltwater provided a certain source of water for the basin, especially during the spring season when rainfall was scarce. Due to such a large area of snow and glaciers, the melting water produces seasonal floods and posed a serious threaten to the downstream residential areas. Through this study, it was able to simulate the hydrological process during flood season in mountain areas and provide technical support for disaster prevention.

**Table 1.** LUCC statistics information table in the study area.

| LUCC | Area ($km^2$) | Percentage (%) |
|---|---|---|
| Glacier and snowfield | 469.96 | 8.35 |
| Bare soil | 815.54 | 14.49 |
| Bare rock | 324.38 | 5.77 |
| Meadow | 1249.17 | 22.20 |
| Sparse grass | 1324.22 | 23.54 |
| River | 69.67 | 1.24 |
| Marsh | 5.65 | 0.10 |
| Evergreen coniferous shrub | 3.98 | 0.07 |
| Grassland | 1202.74 | 21.38 |
| Broadleaved deciduous forest | 3.49 | 0.06 |
| Evergreen needleleaved forest | 138.68 | 2.46 |
| Dryland | 17.42 | 0.31 |
| Settlement place | 1.51 | 0.03 |

*2.3. Methods*

2.3.1. Calculation of Accumulated Temperature

Daily temperatures generally increase initially, then decrease gradually after the maximum value has been reached, with the overall change approximating a sinusoidal curve. The 24 h in a day can be replaced by $0-\pi$, and the daily temperature can be expressed by the formula

$$T_{day} = (T_{mx} - T_{mn}) \sin t + T_{mn} \qquad 0 \le t \le \pi, \tag{1}$$

The accumulated temperature in a day can be calculated by the formula

$$
T = \begin{cases} \int_0^\pi (T_{mx} - T_{mn}) \sin t + T_{mn} \, dt, & 0 \le t \le \pi \\ \int_{\sin^{-1}(\frac{-T_{mn}}{T_{mx}-T_{mn}})}^{\pi-\sin^{-1}(\frac{-T_{mn}}{T_{mx}-T_{mn}})} T_{mx} \sin t \, dt, & \sin^{-1}(\frac{-T_{mn}}{T_{mx}-T_{mn}}), \le t \le \pi - \sin^{-1}(\frac{-T_{mn}}{T_{mx}-T_{mn}}) \end{cases} \tag{2}
$$

where $T_{day}$ is the temperature at any time of the day, $T$ is the daily accumulated temperature, $T_{mx}$ and $T_{mn}$ represent the maximum and minimum daily temperatures, respectively, $t$ is the radian at any time of the day, and $sin^{-1}(\frac{-T_{mn}}{T_{mx}-T_{mn}})$, $\pi - sin^{-1}(\frac{-T_{mn}}{T_{mx}-T_{mn}})$ are the radians corresponding to 0 °C. When the minimum daily temperature was > 0 °C, the first formula in Equation (2) was used to calculate the accumulated temperature; when the minimum temperature was < 0 °C but the maximum temperature was > 0 °C, the second formula in Equation (2) was used [49,50].

The logical approach of this study is shown in Figure 2. Firstly, the precipitation type is determined by expanding the temperature condition in alpine areas and on cold, high mountains. Only when the maximum daily temperature ($T_{max}$) and the accumulated daily temperature ($T_{accu}$) were greater than both the snowfall temperature threshold (SFTMP) and the snowfall accumulated temperature threshold (SFTMP$_{accu}$), the precipitation type is determined as rainfall; otherwise, it is snowfall. The elevation threshold between snow and ice (ETSI) is also added to differentiate snow and ice. The added parameters would improve the performance of the original degree day factor method in more reasonable way. By adding the accumulated temperature threshold (IMTM_A) and the melting temperature threshold (IMTP), ice melt could be simulated by new module. Based on the daily accumulated temperature, a new hourly-scale model simulation was added to evaluate the flood process in mountainous areas. The relationship between long time series meltwater calculation and the NDVI was added in order to evaluate model improvement from an ecological perspective.

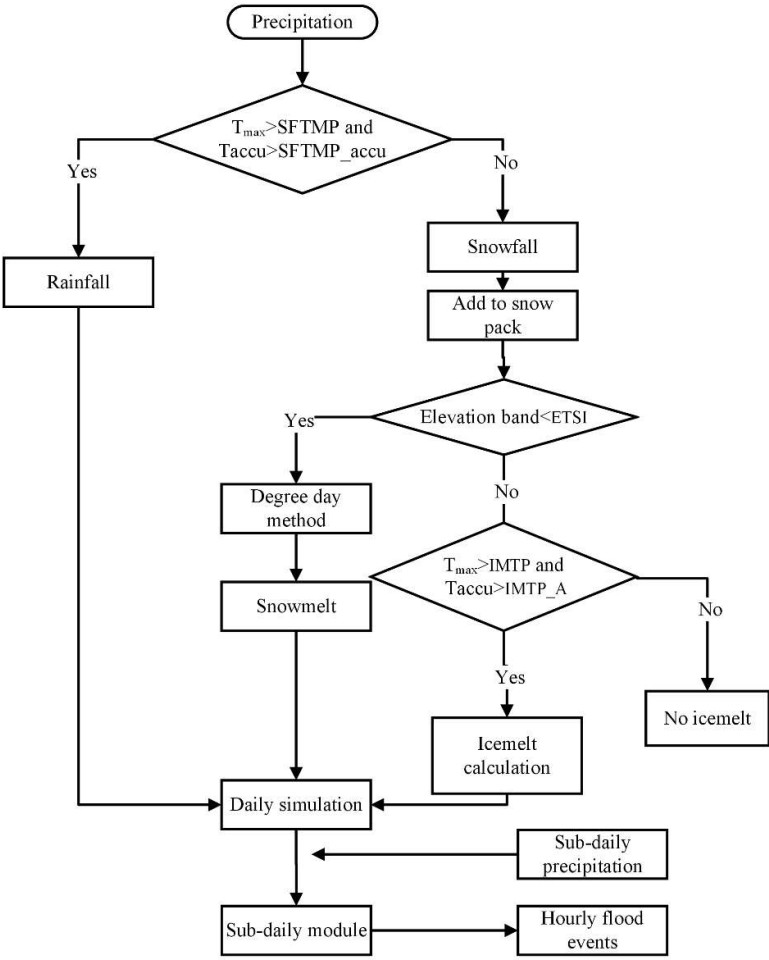

**Figure 2.** Methodological roadmap for this study.

### 2.3.2. Calculation of Ice Melt

Glaciers are often distributed in higher altitudes mountainous areas. Snow cover is usually wider in areas with lower altitudes due to seasonal variations. Therefore, the elevation threshold can be set for segmentation. The ice melt runoff was calculated based on the ice cover condition, and the temperature threshold of the ice melt runoff. The equation for ice melt is

$$ICE_{mlt} = B_{mlt} \cdot ice_{cov} \cdot \left[ \frac{T_{ice} + T_{mx}}{2} - T_{mlt} \right] , \tag{3}$$

where $ICE_{mlt}$ is the amount of ice melt on a given day (mm $H_2O$), $B_{mlt}$ is the melt factor for the day (mm $H_2O$/day-°C), $ice_{cov}$ is the fraction of the hydrological response unit (HRU) area, $T_{ice}$ is the temperature of the ice base, $T_{mx}$ is the maximum air temperature on a given day (°C), and $T_{mlt}$ represents the temperature threshold when the ice melting conditions are reached (°C).

The ice base temperature was an important parameter for the ice melting calculation. Due to temperature hysteresis, the determination of $T_{ice}$ is influenced by conditions during the previous days and varied with temperature. The $T_{ice}$ formula can be expressed as

$$T_{iced_n} = T_{iced_{n-1}} \cdot (1 - \lambda_{ice}) + T_{av} \cdot \lambda_{ice} , \tag{4}$$

$T_{iced_n}$ is the ice base temperature on a given day (°C), $\lambda_{ice}$ is the lagging ice base temperature on the previous day (°C), and $T_{av}$ is the average air temperature on the current day (°C). The most important factor influencing the ice base temperature was the effect of the previous day's temperature on the current day. The lagging factor $\lambda_{ice}$ was included to account for this influence.

The equation for the $B_{mlt}$ calculation is

$$B_{mlt} = \frac{\left(b_{mltJ} + b_{mlt12D}\right)}{2} + \frac{\left(b_{mltJ} - b_{mltD}\right)}{2} \cdot \sin\left( \frac{2\pi}{365} \cdot (d_n - 81) \right), \tag{5}$$

where $b_{mltJ}$ is the melt factor for June 21 (mm $H_2O$/day-°C), $b_{mltD}$ is the melt factor for December 21 (mm $H_2O$/day-°C), and $d_n$ is the numerical day of the year.

Although the new approach of ice melting calculation method originates from the degree day factor method, the added restrictions and new calculation algorithm make the simulation more reasonable. The determinations of precipitation types as well as the temperature condition triggering a melting event are established prior to the calculation of ice melt. This temperature condition was determined by the daily accumulated temperature combined with the maximum temperature. The sub-temperature judgment condition corresponds to the characteristics of 'one day–one peak' flood events and the rapid eruption of ice/snow melt in mountainous areas.

### 2.3.3. Calibration, Validation, and Sensitivity

The sensitivity analyses, calibration, and validation were all indispensable steps of modeling work. In this study, the Soil Water Assessment Tool Calibration and Uncertainty Procedure (SWAT-CUP) [51] was used for the above steps. During the 2011–2014 research period, the first 2 years served as the preheat period of the model, while the years 2013 and 2014 were the calibration and validation periods, respectively. When evaluating model simulation results, the coefficient of determination ($R^2$), the Nash–Sutcliffe efficiency (NSE), and the percent bias (PBIAS) are usually utilized. The calculation formula of $R^2$ is

$$R^2 = \left[ \frac{\sum_{i=1}^{n}(Q_{sim,i} - \overline{Q}_{sim,i})(Q_{obs,i} - \overline{Q}_{obs,i})}{\sqrt{\sum_{i=1}^{n} (Q_{sim,i} - \overline{Q}_{sim,i})^2 \sum_{i=1}^{n}(Q_{obs,i} - \overline{Q}_{obs,i})^2}} \right]^2 , \tag{6}$$

The range of $R^2$ is 0–1; the closer to 1, the better. The NSE ranges from $-\infty$ to 1. Similarly, the closer the NSE is to 1, the more ideal the model results; the further from 1, the worse the results. The calculation formula of the NSE is

$$NSE = 1 - \frac{\sum_{i=1}^{n}\left(Q_{obs,i} - Q_{sim,i}\right)^2}{\sum_{i=1}^{n}\left(Q_{obs,i} - \overline{Q}_{obs,i}\right)^2} \, , \tag{7}$$

The main performance of PBIAS indicators was the state of the simulation of the model compared with the measured values. The positive value of PBIAS indicates that the simulation results of the model are overestimated, while the negative value indicates underestimation. The formula is

$$PBIAS = \frac{\sum_{i=1}^{n}\left(Q_{sim,i} - Q_{obs,i}\right)}{\sum_{i=1}^{n} Q_{obs,i}} 100 \, , \tag{8}$$

The model's performance is rated as either 'very good' (PBIAS < ± 10%), 'good' (± 10% ≤ PBIAS < ± 15%), 'satisfactory' (± 15% ≤ PBIAS < ± 25%), or 'unsatisfactory' (± 25% ≤ PBIAS). The variable $n$ is the total number of the observations. $Q_{obs,i}$ and $Q_{sim,i}$ in the formulas are the $i^{th}$-day observation ($m^3 \cdot s^{-1}$) and the model-simulated discharge ($m^3 \cdot s^{-1}$), respectively; and $\overline{Q}_{sim,i}$ and $\overline{Q}_{obs,i}$ are the average simulation and observation ($m^3 \cdot s^{-1}$), respectively.

Parameter sensitivity analysis can effectively help to determine the optimal combination of model parameters [52]. The sequential uncertainty fitting (SufI-2) algorithm is an important method for sensitivity and uncertainty analyses, in which global sensitivity analysis was the method of partial sensitivity analysis adopted in this study [53,54]. In this method, T-state and *p*-value are important indices in the evaluation of parameter sensitivity. This technique was used to test samples with a T-state hypothesis; the larger the better. The *p*-value is the probability value corresponding to the *t*-test value; the closer the *p*-value of a parameter is to 0, the greater its significance.

## 3. Results

### 3.1. Daily Simulations

The daily-scale simulation results before and after the model modification were compared with the observed discharge, in which the calibration period was studied regularly in 2013 and the validation period was in 2014 (Table 2). Three indices of the model before and after modification were used for the calibration, validation, and overall periods. This comparison revealed that during the calibration period the $R^2$ value increased from 0.8 to 0.87, the NSE increased from 0.73 to 0.77, and the PBIAS improved from 5.42% to 4.55%. During the validation period, all indices of the new SWATAI model are better than the original one, increasing from 0.78% to 0.84%, 0.71% to 0.75%, and −6.89% to 4.85%. During the overall period, the $R^2$, NSE, and PBIAS of comparison results were consistent to those in the calibration and validation periods, with improving 0.05, 0.05, and 3.23%, respectively.

**Table 2.** Evaluation of simulation results before and after model modification.

| Period | $R^2$ | | NSE | | PBIAS (%) | |
|---|---|---|---|---|---|---|
| | SWAT | SWATAI | SWAT | SWATAI | SWAT | SWATAI |
| Calibration (2013) | 0.80 | 0.87 | 0.73 | 0.77 | 5.42 | 4.55 |
| Validation (2014) | 0.78 | 0.84 | 0.71 | 0.75 | −6.89 | 4.85 |
| Overall (2013–2014) | 0.77 | 0.82 | 0.69 | 0.74 | 8.65 | 5.42 |

Each May, the replenishment of river runoff is mainly the result of ice and snow melt. Observations indicate that there was a flood process during this period and the original model could not catch this process (Figure 3). While, the model with new module precisely simulated this flush flood, and the peak value as well as the entire hydrograph were well described. By July and August, the snow at low

elevations has almost disappeared, and the water source of the river mainly comes from ice-melt at high elevations. This period is also the good opportunity to test the performance of new modification. During this period, multiple flood processes were not well simulated by the original SWAT model, because the multiple flood processes were presented as single event and the simulated flood peak was significantly overestimated. While, the SWATAI model can precisely describe the flood process in terms of both timing and value. By the end of August, river runoff has progressively diminished, and there is less ice melt. In the non-snow-melt season after September, the original SWAT model obviously overestimated the recession processes, while the SWATAI model had much better results. The model performance of SWATAI had quite reasonable results in both calibration and validation period.

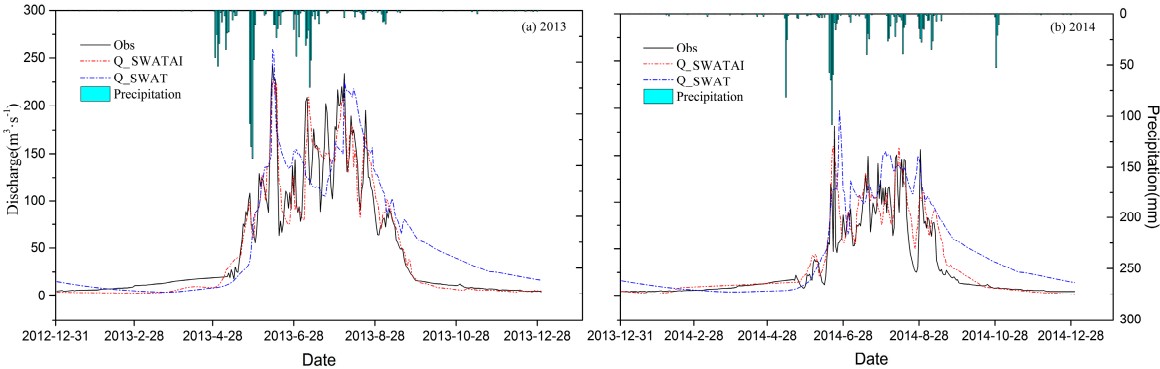

**Figure 3.** Comparison of simulation results from the SWAT and SWATAI models on the diurnal scale.

### 3.2. Sub-Daily Simulations

In order to examine the ice-melt and snow-melt flood processes in more detailed way, a comparative study of the hourly scale flooding process was carried out. During the melting season from May to September, the typical flood events were selected as targets. In Figure 4, it can be seen that the duration of each complete ice/snow-melt flood event last for a relatively short period, generally less than 24 h. If the hydrographs were examined in the aspects of timing and value for each month, the original model produced the lower baseflow, higher and earlier flood peak. This simulation flaw was totally eliminated by the modified model. During the recession period in September, the hydrograph from modified model caught the observation value very well, much better than the original model. Similar outputs were also occurred in the validation period (Figure 5). The original model gave out the results with the significant bias, its flood peaks have larger temporal shifts, overestimation/underestimation, and inappropriate flood duration.

The deviations of flood peak values and timing shifts were statistically summarized for each flood events by the original SWAT model and the SWATAI model. Table 3 indicates that original SWAT model has relatively large errors during the calibration period, the minimum and maximum errors of the peak flood values were $-2.88$ m$^3$·s$^{-1}$ in September, and $63.13$ m$^3$·s$^{-1}$ in June; the minimum and maximum temporal errors were $-1$ hour in August, and 8 h in June. After adding the ice-melt module, the error of the peak flood value in June was reduced to $8.83$ m$^3$·s$^{-1}$, and the temporal error was reduced to $-1$ hour. During the validation period of 2014, the model with new modular also perform better than the original model. The deviations of the flood peak and timing shift in original model were even larger, with a peak value error of $-10.15$ m$^3$·s$^{-1}$ to $96.05$ m$^3$·s$^{-1}$, and a timing shift error $-8$ h to 0 h. By adding the ice-melt module, the peak error shrank to $26.94$ m$^3$·s$^{-1}$, the timing shift was consistently less than 2 h.

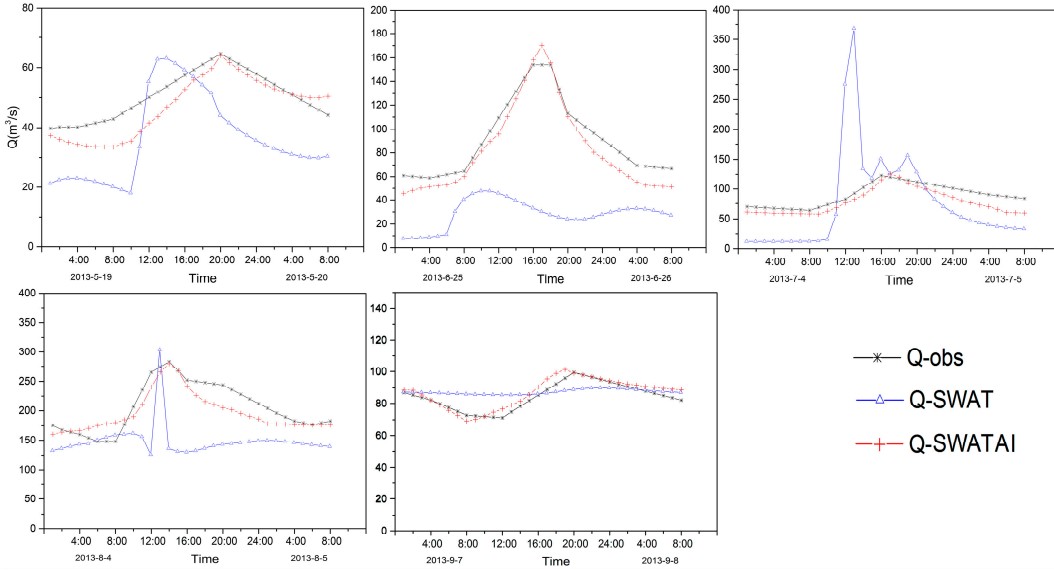

**Figure 4.** Hourly simulation results during the calibration period (2013).

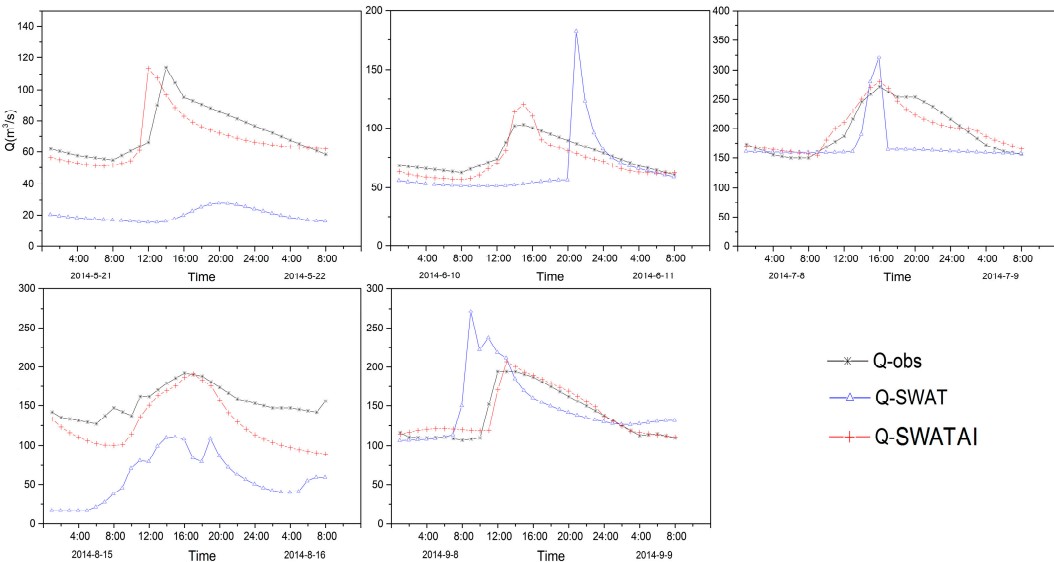

**Figure 5.** Hourly simulation results during the validation period (2014).

**Table 3.** Deviation statistics of peak flood processes before and after model modification.

| Date | Deviation of Flood Peak Value (m³·s⁻¹) | | Deviation of Timing (h) | |
|------|------|------|------|------|
| | **SWAT** | **SWATAI** | **SWAT** | **SWATAI** |
| 2013.5 | 14.52 | 3.88 | 7 | 0 |
| 2013.6 | 63.13 | 8.83 | 8 | −1 |
| 2013.7 | 14.33 | 11.59 | 3 | −1 |
| 2013.8 | 57.05 | 9.7 | −1 | 0 |
| 2013.9 | −2.88 | −2.16 | 2 | 1 |
| 2014.5 | 52.7 | 5.04 | −8 | 2 |
| 2014.6 | 11.23 | 4.37 | −6 | 0 |
| 2014.7 | 26.25 | −2.29 | 0 | 0 |
| 2014.8 | 96.05 | 26.94 | −2 | 1 |
| 2014.9 | −10.15 | −3.03 | 4 | −1 |

### 3.3. Effects of Parameters on the Simulated Results

In order to catch the major processes of ice/snow melt in nature, a new ice-melt module was added to the original model with some essential parameters which can properly describe the nature processes (Table 4). There were a total of six ice-melt–related parameters designed in this study: ice-melt base temperature (IMTP), ice-melt base accumulated temperature (IMTP_A), maximum melt rate for ice during the year (IMFMX), minimum melt rate for ice during the year (IMFMN), elevation threshold between snow and ice (ETSI), and ice temperature lag factor (ITIMP). If the parameters are examined from the temporal point of view, IMTP on the daily and hourly scale was 5.39 °C and was 5.87 °C, while IMTP_A on the daily and hourly scale was 24.68 °C and 26.77 °C Meanwhile, ETSI, IMFMX, and IMFMN did not change appreciably. In the process of model parameter optimization, it was impossible to get the only optimal solution. The general software also gives the parameter range, only through constant calibration, the optimal parameter combination and parameter value can be obtained within the parameter range [55,56].

**Table 4.** Important parameters in the model calibration process.

| File Extension | Parameter | Description | Range of Values | Daily Simulation Calibrated Value | Sub-daily Simulation calibrated Value |
|---|---|---|---|---|---|
| .bsn(New) | ETSI | Elevation threshold between snow and ice | 2000–6000 | 3500 | 3500 |
| .bsn(New) | IMTP | Ice-melt base temperature | −40 | 5.39 | 5.87 |
| .bsn(New) | IMTP_A | Ice-melt base accumulated temperature | 0–40 | 24.68 | 26.77 |
| .bsn(New) | IMFMX | Maximum melt rate for ice during the year | 0–20 | 12.35 | 13.04 |
| .bsn(New) | IMFMN | Minimum melt rate for ice during the year | 0–20 | 15.87 | 16.55 |
| .bsn(New) | ITIMP | Ice temperature lag factor | 0–1 | 0.61 | 0.65 |
| .bsn(New) | SFTMP_A | Snowfall accumulated temperature | 0–40 | 28 | 29 |
| .bsn | SFTMP | Snowfall temperature | −20 to 20 | 3.27 | 3.46 |
| .bsn | SMTMP | Snow-melt base temperature | −20 to 20 | 3.06 | 2.85 |
| .bsn | SMFMX | Maximum melt rate for snow during the year | 0–20 | 7.62 | 7.57 |
| .bsn | SMFMN | Minimum melt rate for snow during the year | 0–20 | 9.4 | 8.19 |
| .bsn | TIMP | Snowpack temperature lag factor | 0–1 | 0.55 | 0.54 |
| .bsn | SNOCOVMX | Minimum snow water content corresponding to 100% snow cover | 0–500 | 38.33 | 37.84 |
| .bsn | SFTMP | Snowfall temperature | −40 | 3.36 | 3.47 |
| .bsn | SURLAG | Surface runoff lag time | 0.05–24 | 11.78 | 11.43 |
| .gw | ALPHA_BF | Base flow alpha factor (days) | 0–1 | 0.15 | 0.18 |
| .gw | GW_DELAY | Groundwater delay (days) | 0–500 | 222.68 | 224.12 |
| .gw | GWQMN | Threshold water depth in the shallow aquifer required for return flow to occur (mm) | 0–5000 | 1175.84 | 1205.64 |
| .gw | SHALLST | Initial water depth in the shallow aquifer (mm) | 0–50,000 | 4903.68 | 4958.74 |
| .gw | GW_REVAP | Groundwater "revamp" coefficient | 0.02–0.2 | 0.06 | 0.05 |
| .mgt | CN2 | SCS runoff curve number | 35–98 | 70.79 | 72.35 |
| .ohru | OV_N | Manning's "n" value for overland flow | 0.01–30 | 10.77 | 11.12 |
| .ohru | ESCO | Soil evaporation compensation factor | 0–1 | 0.37 | 0.35 |
| .ohru | EPCO | Plant uptake compensation factor | 0–1 | 0.39 | 0.32 |
| .rte | CH_N2 | Manning's "n" value for the main channel | −0.01 to 0.3 | 0.01 | 0.01 |
| .rte | CH_K2 | Effective hydraulic conductivity in main channel alluvium | −0.01 to 500 | 47.38 | 48.12 |
| .sol | SOL_K | Saturated hydraulic conductivity | 0–2000 | 861.31 | 874.58 |
| .sol | SOL_AWC | Available water capacity of the soil layer | 0–1 | 0.32 | 0.36 |
| .sub | PLAPS | Precipitation lapse rate | −20 to 20 | −5.5 | −5.36 |
| .sub | TLAPS | Temperature lapse rate | −10 to 10 | −7.59 | −7.64 |
| .sub | CH_N1 | Manning's "n" value for the tributary channels | 0.01–30 | 5.42 | 5.13 |
| .sub | CH_K1 | Effective hydraulic conductivity in tributary channel alluvium | 0–300 | 295.67 | 271.36 |
| .sub | SNO_SUB | Initial snow water content | 0–150 | 95.39 | 97.33 |

The sensitivity values of 32 parameters involved in model calibration at both daily and hourly scales were analyzed. As can be seen in Table 5, CH_K2 was the most sensitive parameter, with a T-state value of 21.67 and a $p$-value of 0. PLAPS was also highly sensitive, with a T-state value of 18.53 and a $p$-value of 0. The newly-added ice-melt parameter IMTP_A had a T-state value of 12.06 and a $p$-value of 0, indicating high sensitivity. The T-state value of IMTP was 9.05, and its $p$-value was 0. The T-state value of the ETSI parameter was 8.25, and its $p$-value was 0.01, also revealing high sensitivity. Although the sensitivity levels of IMFMX and IMFMN were slightly lower, they displayed a certain sensitivity as well. In addition, relevant parameters involved in snow-melt and ice-melt calculations, such as SMTMP, TLAPS, SMFMX, and SFTMP, all exhibited a certain amount of sensitivity.

**Table 5.** Parameter information for global sensitivity.

| Parameter | T-states | *p*-Value |
|---|---|---|
| CH_K2 | 21.67 | 0 |
| PLAPS | 18.53 | 0 |
| IMTP_A | 12.06 | 0 |
| SMTMP | 10.98 | 0.01 |
| TLAPS | 9.76 | 0.01 |
| IMTP | 9.05 | 0.01 |
| ETSI | 8.25 | 0.01 |
| LAT_TTIME | 6.89 | 0.02 |
| SMFMX | 6.05 | 0.02 |
| SOL_K | 5.36 | 0.03 |
| IMFMX | 5.09 | 0.03 |
| SOL_AWC | 4.68 | 0.05 |
| IMFMN | 4.47 | 0.05 |
| SURLAG | 3.28 | 0.06 |
| TIMP | 2.64 | 0.17 |
| ITIMP | 2.36 | 0.18 |
| GWQMN | 1.49 | 0.26 |
| SNO_SUB | 1.05 | 0.34 |
| REVAPMN | 1.01 | 0.57 |
| EPCO | 0.72 | 0.64 |
| SMFMN | 0.43 | 0.79 |
| CH_N2 | 0.38 | 0.82 |
| RCHRG_DP(Deep aquifer percolation fraction) | 0.21 | 0.89 |
| OV_N | −0.02 | 0.91 |
| CH_N1 | −0.57 | 0.71 |
| SNOCOVMX | −1.05 | 0.53 |
| SHALLST | −1.24 | 0.51 |
| ALPHA_BF | −2.49 | 0.43 |
| CN2 | −3.07 | 0.39 |
| CH_K1 | −3.14 | 0.34 |
| SFTMP | −3.88 | 0.26 |
| GW_DELAY | −4.01 | 0.14 |

### 3.4. Relationship between NDVI and Model Modification

The snow and ice melting in the arid and semi-arid areas are of great significance to the local ecology and production, including the green vegetation in spring and the crop planting in the lower reaches. Snow and ice melting directly affect the vegetation growth. While, NDVI can reflect the growth of vegetation in spring. In this study, the correlation between vegetation NDVI and the amount of melting water can indirectly verify the effect of the model modification and the accuracy of the calculation from an ecological perspective. The vegetation variation in the study area as reflected by the NDVI can be used to evaluate the modification effect of the model. In this study, MODIS Surface Reflectance MOD09Q1 data from 2000 to 2015 were used to calculate the NDVI. The meteorological data used in the model were derived from ERA-Interim meteorological data. The relationships between the NDVIs of different months in spring were shown in Figure 6, and the amount of melt water were compared on the daily time scale. A comparison of the NDVIs of the spring months in the study area with the river recharge area reveals that while the vegetation began turning green in March, the maximum NDVI value in the area did not exceed 0.71, and the vegetation area was primarily concentrated in the recharge area of the lower reaches of the river. In April, the maximum NDVI value was 0.92, and the vegetation area had further expanded. In May, the maximum NDVI value was 0.99, and the vegetation area had continued to expand, with the vegetation area in the upstream region of the study area increasing significantly.

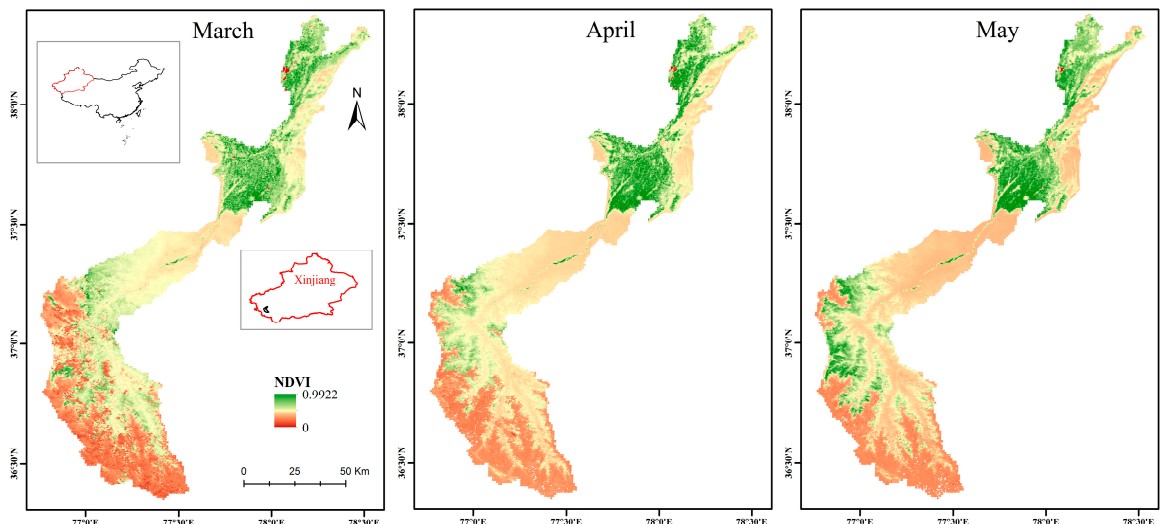

**Figure 6.** Springtime NDVI distribution in the research area.

The response of the relationship between the NDVI and the amount of snow and ice melt calculated by the models was analyzed. Three types of land use were selected: herbaceous green space, dry land, and paddy fields. As shown in Figure 7a, there were a large number of 0 values for snow and ice melt calculated by the SWAT model in spring, which was inconsistent with the actual situation. The correlation coefficient calculated between snow and ice melt with the NDVI was only 0.58. In Figure 7b, there were no 0 values for snow and ice melt calculated by the SWATAI model, and the correlation coefficient increased to 0.76. For the paddy fields, the correlation coefficient between the NDVI and snow and ice melt was 0.65 for original SWAT model, and after the ice-melt module was added, the correlation coefficient increased to 0.77 (Figure 8). As shown in Figure 9, the correlation coefficient between the snow and ice melt calculated by the model and the dry land NDVI increased from 0.67 to 0.80 after model modification.

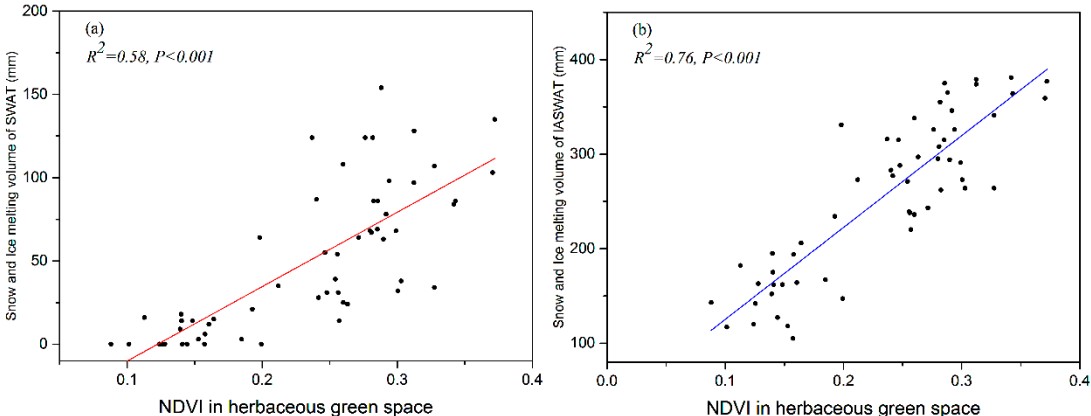

**Figure 7.** Relationship between the herbaceous green space NDVI and snow and ice melt: (**a**) relationship between the amount of snow and ice melt calculated by the SWAT model and the herbaceous green space NDVI; (**b**) relationship between the amount of snow and ice melt calculated by the SWATAI model and the herbaceous green space NDVI.

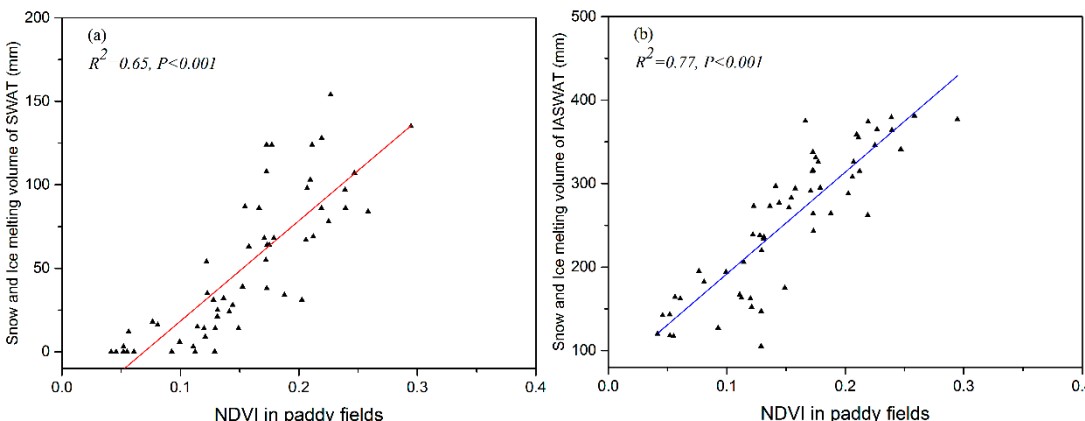

**Figure 8.** Relationship between the paddy field NDVI and snow and ice melt: (**a**) relationship between the amount of snow and ice melt calculated by the SWAT model and the paddy field NDVI; (**b**) relationship between the amount of snow and ice melt calculated by the SWATAI model and paddy field NDVI.

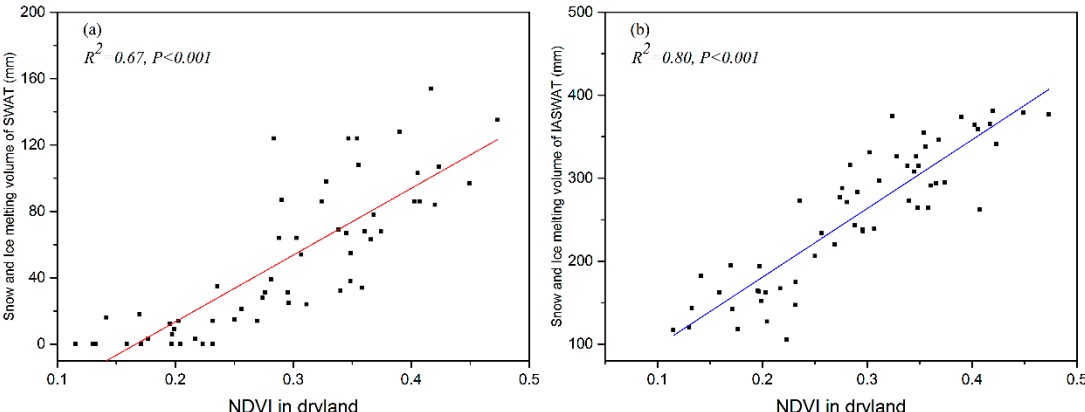

**Figure 9.** Relationship between the dry land NDVI and snow and ice melt: (**a**) relationship between the amount of snow and ice melt calculated by the SWAT model and the dry land NDVI; (**b**) relationship between the amount of snow and ice melt calculated by the SWATAI model and the dry land NDVI.

## 4. Discussion

### 4.1. Model Modification

Snow and ice are not clearly distinguished in the SWAT model [57]. Thus, this method is obviously not applicable in alpine mountainous areas, and its calculation of ice and snow melt deviates from observation. In this study, the SWATAI model was based on the original SWAT model, with the addition of a new ice-melt module, which was developed to overcome the deficiency of the model's ice-melt calculation. A newly-added ice-melt module refers to the calculation method of snow and the enhanced relevant parameters of ice-melt. Temperature, as an important parameter in model calculation, is also an important factor affecting the process of ice and snow melt [42,58]. Temperature is mainly reflected in the determination of precipitation type and the important conditions affecting the determination as to whether ice or snow melting occurs [51,58,59]. Therefore, the SWATAI model mainly considered the modification of the model in terms of temperature and the accurate differentiation and calculation of snow and ice in the model. Snow and ice were distinguished by adding an elevation threshold. When the elevation was less than the elevation threshold, the precipitation was mainly considered to be snow; otherwise, it was ice. In addition, as an important temperature discrimination condition, the average daily temperature was utilized in the traditional degree day factor method but was not applicable in

the high and cold mountainous areas. Therefore, the maximum temperature and the accumulated temperature were added as the temperature determination conditions in the new ice-melt module.

The daily temperature integral method was applied for the calculation of accumulated temperature. In order to improve the recognition accuracy of the precipitation types and to avoid the occurrence of accidental events, the maximum daily temperature threshold was also increased. As a result, the model with the ice-melt module has higher accuracy and applicability in the simulation of snow melt and ice melt floods in mountainous regions, at both the daily and hourly time scales. The SWATAI model can choose different simulation scales according to different research purposes. In the context of future climate change, the annual scale can be used to analyze the relationship between snow, glacier area, and runoff. In the research of flood disaster, hourly scale is a better choice. The daily scale and monthly scale can be used in the research of water resources management. research on water resources management can use daily and monthly scales. The daily scale and weekly scale can be used in the water resources management for reservoirs and agricultural water.

## 4.2. Model Performance Comparison

Figures 3–5 illustrate the comparison between the simulated discharges and observations on the daily and hourly time scales, respectively, and were created to more intuitively analyze the modification effect of the SWATAI model. Prior to April, there was little difference between simulated runoff and observations. This was mainly due to low temperatures from January through March, resulting in no melting of the upstream snow and glaciers [60,61]. In April, temperature began to rise, and the snow and ice at lower altitudes began to melt. There was a flood event in May, to which the original SWAT model did not respond, while the SWATAI model simulated it well. This difference between the model simulations was due to the fact that the surface accumulated temperature, which is utilized in the SWATAI model, is more sensitive than the air temperature, especially in areas without vegetation cover at high altitudes. This modification avoids neglecting events for which the daily average temperature does not reach the temperature threshold in the original model, although ice-melt events still do occur in this model [22]. In June, river runoff increases significantly, mainly due to rapid temperature rise and the associated acceleration of snow and ice melt. In general, flooding processes in summer were overestimated by the original SWAT model. In addition, multiple flood processes were combined and presented as one flooding process. This was mainly due to the fact that the original model only uses the daily average temperature to determine the conditions of ice-melt, thus neglecting the large diurnal temperature range in the mountains. At night, temperatures in the mountains decrease rapidly, and the ice-melting process stops [22]. These details cannot be taken into account using the daily average temperature alone. However, when the accumulated temperature determination conditions were added, the diurnal temperature range was fully considered [62]. Therefore, each flood process could be well simulated, and overestimation by the model calculations could be avoided to a certain extent, thereby solving the inability to distinguish individual flood processes [63]. In September, temperatures in the mountains drop sharply, and the melting process of snow and ice gradually weakens and disappears [10]. The simulation discharges of the original SWAT model were obviously larger than the observations, mainly due to the lack of a method for distinguishing snow and ice and calculating them separately. Once the snow at low elevations has disappeared, ice still remains at higher elevations. If the ice-melt is still calculated according to the threshold and conditions set by the snow-melt, it will definitely be overestimated, since the temperatures at this time may not reach the temperature threshold of ice-melt [64].

In order to better analyze and study the mountain flooding process and to verify the simulation accuracy of the new model, it was necessary to examine the flood process on the hourly scale [50,59]. 10 flood events during the 2-year study period were selected for statistical analysis, the maximum time deviation of the simulation results of the original SWAT model was 8 h, which obscures the significance of studying flood processes at high time resolution. In the simulation results of the SWATAI model, the time deviation was less than 2 h, and the accuracy was greatly improved. One of the implications

of the model modification was that it can perfectly represent the entire process of flood expansion and regression. Based on the simulation results of the original SWAT model, the complete process could not be well represented, and the peak value deviation was as much as 96.05 $m^3 \cdot s^{-1}$. This was mainly due to the calculation of melting ice and snow as a single process [4]. The peak flood values simulated by the models of all 25 flood events during the calibration and validation periods were statistically analyzed and compared to the measured data (Figure 10). The scatter plot clearly indicates that the simulated results of the model with the added ice-melt module were closer to the measured values over the two periods. In order to compare the correlation between the simulated and measured flood peaks before and after the model modification, R-squared values were calculated for the 25 flood peaks during the calibration and validation periods. By calculation, the R-squared value between the simulated peaks of the original model and the measured peaks was 0.85 during the calibration period, and that was 0.9 in the modified results. During the verification period, the correlation between the peak value of the original model and the measured data was 0.83, which increased to 0.89 by modifying the model. Through the modification, whether in the calibration period or validation period, the simulated value of the modified model has a higher correlation with the observed data than the simulated values. This was mainly because through the modification of the model, the simulation of the flood peaks was closer to the measured data, which improved the phenomenon that the original model generally underestimated the flood peaks. These results also confirm the rationality and importance of the addition of the ice-melt module and provide a better exploration of flooding process simulation.

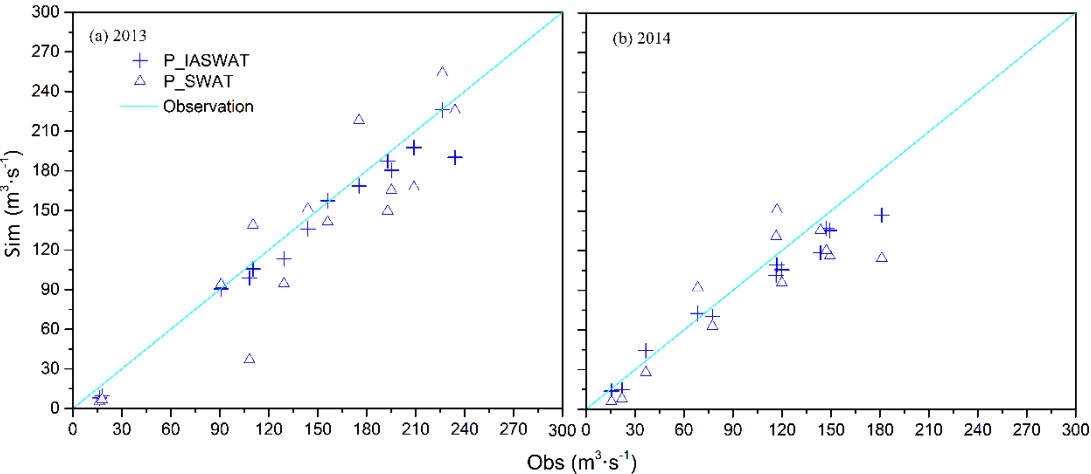

**Figure 10.** Comparison between the simulated flood peak values and the measured data of all 25 flood events during the model calibration and validation periods.

### 4.3. Analysis of Parameter Sensitivity and Uncertainty

The analysis of sensitivity and uncertainty provides indispensable and important information in model-related research [52]. During the model calibration process, the sensitivity of each parameter is analyzed via continuous calibration and screening in order to determine the optimal combination of parameters [57]. The results of the parameter sensitivity analysis in this study are listed in Table 4. The effective hydraulic conductivity in the alluvium of the main channel (CH_K2) was the most sensitive of all the parameters, followed by precipitation lapse rate (PLAPS) and ice-melt base accumulated temperature (IMTP_A). Snow-melt base temperature (SMTMP), temperature lapse rate (TLAPS), ice-melt base temperature (IMTP), and the elevation threshold between snow and ice (ETSI) also exhibited high sensitivity. As important parameters of the newly-added ice-melt module, all of these parameters displayed high sensitivity, thus indicating to a certain extent the rationality of parameter performance after the addition of the ice-melt module. Due to its complex terrain and large elevation variation, the Tizinafu River Basin has an extensive area covered by snow and glaciers. During the process of model calibration, parameters related to snow and ice melt, as well as elevation, are

highly sensitive, which was indeed verified by the statistical results of the parameter sensitivity analysis [51,54]. The sensitivity analysis of the parameters makes it possible to better filter the parameters and to improve the calculation accuracy of the model [52,53]. When analyzing the model uncertainty, the structure, data, and parameters of the model were regularly evaluated [65], allowing the simulation results to reach their optimal values for the given conditions, although uncertainty was inevitable [65]. Indeed, when analyzed the parameter sensitivity of the model, all the parameters used in the model calibration were counted. While, most parameters related other hydrological processes, such as evapotranspiration, infiltration, surface roughness, could significantly influence the model output in very sensitive way. If come to these specified processes, such as snow melting or ice melting, it has to focus on the newly added module of melting and its parameters in order to describe the melting process more reasonably. A separate ice melting calculation module and some new parameters were added, including the elevation threshold of snow and glacier boundary (ETSI), the temperature threshold of ice melting (IMTP), the accumulated temperature threshold (IMTP_A), the maximum ice melting factor (IMFMX), and the minimum ice melting factor (IMFMN), the ice melting temperature delay parameter(ITIMP), as well as the snowfall accumulated temperature threshold (SFTMP_A). Uncertainty mainly comes from the input data, model structure, and parameter [55,66]. When reducing the uncertainty of the input data, the model input data was usually compared and screened to remove outliers [67,68]. The most accurate and suitable data was chosen and verified to a certain extent. When reducing the uncertainty of the parameters, we usually perform repeated calibration with the help of the calibration tool (SUFI-2) to ensure the optimal parameter value and parameter combination [69–71]. As for the uncertainty of the model structure, it can only be found in practical applications, and the model structure is continuously improved. In this study, through the distinction between snow melting and ice melting calculation, a new ice melting calculation module was added, the structure of the model was optimized, and the simulation results were more accurate. Compared with the original model, the uncertainty of the model structure has been improved.

### 4.4. Relationship Between NDVI and Snow and Ice Melt in Spring

The Tizinafu River flows into Yecheng County through the Jiangka hydrology station and provides water for agricultural irrigation. In the spring, the snow and ice melt provide sufficient water for herbaceous vegetation in the mountain areas [72–74]. The downstream region is used for farming in the spring, and river water is employed for agricultural irrigation. Therefore, in Figure 6, the NDVI in March was mainly concentrated in the downstream irrigation areas, and the green vegetation was only found in a few mountainous regions. Natural vegetation was scarce in the downstream region, which is primarily an irrigation area where certain crops are grown. In April, as the temperature gradually increased, snow and ice melt accelerated, the regeneration of mountain herbs accelerated, the NDVI value gradually increased, and the vegetation area continued to expand [64,75,76]. The mountainous vegetation area and the downstream irrigation zone both expanded through the use of river water [77]. In May, the snow and ice in the mountainous areas continued to melt, vegetation grew rapidly, and the vegetation area continued to increase. The vegetation in the downstream irrigation areas and residential areas grew well. The herbaceous green land in the mountainous areas was the vegetation that began to turn green in early spring, which may well represent the rapid feedback of mountain vegetation to water supply. Paddy fields and dry land were the two main kinds of local cropland. During the dry spring, farmers in the Kunlun Mountains region primarily rely on river water for irrigation. Thus, in the spring, agriculture is carried out using river water sources, and the two land types may well represent the response of vegetation in downstream areas to the amount of snow and ice melt. From Figure 6, it can be seen that the NDVI values and vegetation area in March, when there was less melting water, were significantly smaller than those in April. In May, along with temperature, water supply also played an important role [78]. This was mainly due to the fact that the original model only calculated the ice as snow cover. In fact, ice melting lasted longer and the amount of water stored was larger, leading to a large error in the calculation of the continuity of water melting in the

original SWAT model, resulting in the absence of meltwater during some periods [78]. The drylands and paddy fields in the study area were mainly distributed in the lower reaches. In winter, there is lack of water resources. When the temperature rises in spring, the snow and ice melt quickly recharged the river, and the runoff increased continuously. The farmlands mainly depend irrigation water due to the sparse rainfall in the downstream area. Winter wheat and walnut were the main crop types in dryland, and rice was the major crop in paddy field [79,80]. In spring, when winter wheat was irrigated with river water, the wheat will turn green rapidly. Hereby, NDVI value in dryland has a strong sensitivity to the stream flow, or in other words, melting water. The reforestation of walnuts and other crops was also concentrated in April. While the growing season of rice in paddy field begins in late May with huge water demand. Therefore, NDVI of dryland field vegetation was highly correlated with the amount of melting water. The NDVI and melting water from March to May in spring were used for correlation analysis. During this period, the growth of rice was very slow, and the effect of water storage in paddy field has a certain delay on the response of melting water, which resulted in the correlation between NDVI and melting water in paddy field was slightly lower than that in dry land. As a result of the modification, the correlation between the amount of snow and ice melt calculated by the model and the NDVI has been greatly improved.

## 5. Conclusions

A SWATAI model with the independent ice-melt module was developed based on the original SWAT model in which referenced the degree day factor method and also utilized new temperature determination conditions. The daily scale accuracy of the new SWATAI model was much higher than that of the original SWAT model. In particular, during the high incidence of ice melt floods in summer, multiple flood processes were considered to be a single event by the original SWAT model—thus effectively ignoring the growth and regression of individual processes—were well simulated by the SWATAI model. In the comparison of the simulation results, the SWATAI model increased the $R^2$ factor by 7.7% compared to the original SWAT model, the NSE value increased by 6.1%, and the PBIAS increased by 29.3%. In addition, the contribution rate of snow and ice melt in the river discharge increased by 18.73%. A comparison of the peak value and temporal deviations of all flood events during the flood season revealed that the accuracy of the SWATAI model in terms of peak deviation was improved by 77.65%, while the time deviation accuracy was improved by 82.93%.

The meltwater volume was calculated using the long-term sequence of meteorological data such as precipitation and temperature and compared to NDVI data in order to verify the accuracy of the model simulation results. It was determined that the $R^2$ value between the NDVI and the amount of meltwater increased from 0.58 to 0.76 in herbaceous green spaces, from 0.65 to 0.77 in paddy fields, and from 0.67 to 0.80 for dry land.

This study provided a new method for simulating the multiple flood processes with high temporal resolution in alpine mountainous areas and presented a new benchmark for the flood generation mechanism as well as early warning and prediction research in mountainous areas. These tools and results can provide technical support and serve as an analysis reference for flood disaster prevention and economic benefit assessment.

**Author Contributions:** Conceptualization, Y.D., T.L., and Y.H.; Methodology, Y.D. and T.L.; Software, Y.D. and F.M.; Validation, Y.D., M.L., and W.X.; Investigation, Y.D.; Resources, T.L. and Y.H.; Data curation, Y.D., Y.Y., and M.L.; Writing—original draft preparation, Y.D.; Writing—review and editing, F.M., V.N., and P.D.M.; Supervision, T.L., Y.H., and P.D.M.; Project administration, W.X.; Funding acquisition, T.L. and Y.H. All authors have read and agreed to the published version of the manuscript.

**Funding:** This research was funded by the International Cooperation Project of the National Natural Science Foundation of China ,41761144079; the Strategic Priority Research Program of the Chinese Academy of Sciences, XDA20060303; the State's Key Project of Research and Development Plan of China, 2017YFC0404501; the International Partnership Program of the Chinese Academy of Sciences, 131551KYSB20160002 and the Project of Research Center for Ecology and Environment of Central Asia of CAS, Y934031.

**Acknowledgments:** The authors would like to express their gratitude to the China Meteorological Data Sharing Service System and Xinjiang Tarim River Basin Management Bureau for providing the climate and river flow data, respectively. We are very appreciative of the Key Laboratory of Remote Sensing and Geographic Information System, Xinjiang Institute of Ecology and Geography for providing the LUCC data and the Food and Agriculture Organization (FAO) for providing the soil map. We thank LetPub (www.letpub.com) for its linguistic assistance during the preparation of this manuscript.

**Conflicts of Interest:** The authors declare no conflict of interest.

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
