# Peer review of "Accurate Simulation of Ice and Snow Runoff for the Mountainous Terrain of the Kunlun Mountains, China"

_remotesensing, doi:10.3390/rs12010179_

Round 1

Reviewer 1 Report

The manuscript “Accurate simulation of ice and snow runoff for the mountainous terrain of the Kunlun Mountains, China” by Duan et al. is a nice try to promote the SWAT modeling results in alpine environments by improve some ice and snowfall related parameters. However, there are some issues need to be further clarified before published.

Better to provide the statistics of land use/cover in the Tizinafu River Basin. So the audience could know in what extent the author’s effort potentially improve the model simulations. The model sensitivity analyses showed a few parameters were sensitive, but the authors select single one parameter (not the most sensitive one) for model improvement, why? The authors select the ETSI, IMTM_A and other parameters for their improvement. Since many calibration and validation tools have already developed for model runs, people always can find a matrix of parameter combination to promote the model performance. So, the key issue is how to verify the parameters are the true values, or response to the true environment? It is a nice try to combine the remote sensing data into the model, however, the authors need detailed explanation for how integrate the NDVI into the model simulation, given that the spatial and temporal scale of them are totally different.

Author Response

Point-by-Point Response to Reviewers (version of the draft)

On the manuscript “Accurate simulation of ice and snow runoff for the mountainous terrain of the Kunlun Mountains, China” by Yongchao Duan, Tie Liu, Fanhao Meng, Ye Yuan, Min Luo, Yue Huang, Wei Xing, Vincent NZABARINDA, Philippe De Maeyer,

Dear reviewers and editor:

Thank you very much for your valuable comments and kind suggestions on our submission. Your academic sense and scientific literacy greatly promoted this manuscript. We highly appreciate your time and efforts. The manuscript has been modified according to your comments. The detailed responses are followed in below point by point.

Responses to the comments of the reviewer

Point 1: Better to provide the statistics of land use/cover in the Tizinafu River Basin. So the audience could know in what extent the author’s effort potentially improve the model simulations.

Response 1:

Thank the reviewer for this suggestion. Indeed, the statistic information could help the audiences to understand the background information of the study region. We have added the new table of land use/cover symbols and the related describe. (line 157-167, page 5)

“The classification and statistics of the land use and land cover (Table 1) shows that the glacier and snowfield take considerable portion of study area. The area of glacier and snowfield in the study region was as high as 469.96 km2, accounting for 8.35% of the total area. Snow and glacial meltwater provided a certain source of water for the basin, especially during the spring season when rainfall was scarce. Due to such a large area of snow and glaciers, the melting water produces seasonal floods and posed a serious threaten to the downstream residential areas. Through this study, it was able to simulate the hydrological process during flood season in mountain areas, and provide technical support for disaster prevention.”

Table 1. The LUCC statistics information table in the study area.

LUCC

Area(km2)

Percentage(%)

Glacier and snowfield

469.96

8.35

Bare soil

815.54

14.49

Bare rock

324.38

5.77

Meadow

1249.17

22.20

Sparse grass

1324.22

23.54

River

69.67

1.24

Marsh

5.65

0.10

Evergreen coniferous shrub

3.98

0.07

Grassland

1202.74

21.38

Broadleaved deciduous forest

3.49

0.06

Evergreen needleleaved forest

138.68

2.46

Dryland

17.42

0.31

Settlement place

1.51

0.03

Point 2: The model sensitivity analyses showed a few parameters were sensitive, but the authors select single one parameter (not the most sensitive one) for model improvement, why?

Response 2:

Thanks for posing this question. We apologize for unclear description for parameter selection. (line 537-547, page 20)

“Indeed, when analyzed the parameter sensitivity of the model, all the parameters used in the model calibration were counted. While, most parameters related other hydrological processes, such as evapotranspiration, infiltration, surface roughness, could significantly influence the model output in very sensitive way. If come to these specified processes, such as snow melting or ice melting, it has to be focused the newly added module of melting and its parameters in order to describe the melting process more reasonable. A separate ice melting calculation module and some new parameters were added, including the elevation threshold of snow and glacier boundary (ETSI), the temperature threshold of ice melting (IMTP), the accumulated temperature threshold (IMTP_A), the maximum ice melting factor(IMFMX) and the minimum ice melting factor(IMFMN), the ice melting temperature delay parameter(ITIMP) as well as the snowfall accumulated temperature threshold (SFTMP_A).”

Point 3: “The authors select the ETSI, IMTM_A and other parameters for their improvement. Since many calibration and validation tools have already developed for model runs, people always can find a matrix of parameter combination to promote the model performance. So, the key issue is how to verify the parameters are the true values, or response to the true environment?

Response 3:

Thank reviewer for the good question. This issue is surely you mentioned is also an important issue that is often concerned and mentioned in model calibration.

“In the process of model parameter optimization, it was impossible to get the only optimal solution. The general software also gives the parameter range, only through constant calibration, the optimal parameter combination and parameter value can be obtained within the parameter range[55,56].” (line 357-360, page 12)

Point 4: “It is a nice try to combine the remote sensing data into the model, however, the authors need detailed explanation for how integrate the NDVI into the model simulation, given that the spatial and temporal scale of them are totally different.

Response 4:

Thank reviewer for the suggestion. We apologize for the insufficient description.  “The snow and ice melting in the arid and semi-arid areas are of great significance to the local ecology and production, including the green vegetation in spring and the crop planting in the lower reaches. Snow and ice melting directly affect the vegetation growth. While, NDVI can reflect the growth of vegetation in spring. In this study, the correlation between vegetation NDVI and the amount of melting water can indirectly verify the effect of the model modification and the accuracy of the calculation from an ecological perspective.” (line374-379, page 15).

Reviewer 2 Report

Thank you so much for providing an opportunity to read this wonderful work by Duan et al. In general, the paper is good. Addition of literature on improvisation of SWAT model and its benefits, and its application to understand mountain hydrological process in Introduction section would make the manuscript more relevant to wider audiences.

Author Response

Point-by-Point Response to Reviewers (version of the draft)

On the manuscript “Accurate simulation of ice and snow runoff for the mountainous terrain of the Kunlun Mountains, China” by Yongchao Duan, Tie Liu, Fanhao Meng, Ye Yuan, Min Luo, Yue Huang, Wei Xing, Vincent NZABARINDA, Philippe De Maeyer,

Dear reviewers and editor:

Thank you very much for your valuable comments and kind suggestions on our submission. Your academic sense and scientific literacy greatly promoted this manuscript. We highly appreciate your time and efforts. The manuscript has been modified according to your comments. The detailed responses are followed in below point by point.

Responses to the comments of the reviewer

Point 1: Addition of literature on improvisation of SWAT model and its benefits, and its application to understand mountain hydrological process in Introduction section would make the manuscript more relevant to wider audiences.

Response 1:

Thank the reviewer for this suggestion. Relevant content has been added according to your suggestion.

“Some researchers proposed an improved SWAT-based computational framework of critical source areas identification at the lake basin scale, which improved the ability of supplying a more comprehensive delineation of critical source areas[44]. In the studyies of the SWAT model in the process of mountainous runoff, some scholars have tried to evaluate the impacts of temperature and precipitation changes on runoff and streamflow by trying to add ice melting algorithms[4,47,48].”(line78-87,page2)

Reviewer 3 Report

The paper is well organized, and the ice-melt module was added to SWAT model, which is innovative to some extent.Based on the modification, the accuracy of the daily flow simulation results has been considerably improved. By employing a detailed comparison between the normalized difference vegetation index (NDVI) and the quantity of meltwater, the model calculation results can be verified, and the impact of meltwater on the ecology can be highlighted.
This paper has certain application value. I recommend acceptance for publication.

Author Response

Dear reviewer:

We thank you very much for your affirmation of our work, which is also the driving force for us to keep moving forward and continue relevant research. Thank you again!

Reviewer 4 Report

I appreciate the Editor to give me a chance to review an interesting and valuable paper. I found some merits in the both methodology and results. In my opinion, this paper has a good potential to be published in the journal. However, I have also some concerns on the different parts of the manuscript. If the author(s) address carefully to the comments, I’ll recommend publication of the manuscript in the journal:

Add/Replace the name of the study area to the Keywords. In the last paragraph of the Introduction, the authors should clearly mention the weakness point of former works (identification of the gaps) and describe the novelties of the current investigation to justify us the paper deserves to be published in this journal. In the Tables, highlight values that are more important and discuss them for better understanding readers. Discuss the reasons of the better performance of the models for drylands compared to the paddy fields. Add R-squared values for comparison between the simulated flood peak values and the measured data of all 25 flood events during the model calibration and validation periods and discuss on the values obtained. What are the strategies/recommendations to reduce uncertainties in this study? Who can we use the SWATAI model for long term scales (i.e. daily, weekly, monthly, annual)? How can extend the results in other regions with similar/different climates? The quality of the language needs to improve by a native English speaker for grammatically style and word use.

Author Response

Point-by-Point Response to Reviewers (version of the draft)

On the manuscript “Accurate simulation of ice and snow runoff for the mountainous terrain of the Kunlun Mountains, China” by Yongchao Duan, Tie Liu, Fanhao Meng, Ye Yuan, Min Luo, Yue Huang, Wei Xing, Vincent NZABARINDA, Philippe De Maeyer,

Dear reviewers and editor:

Thank you very much for your valuable comments and kind suggestions on our submission. Your academic sense and scientific literacy greatly promoted this manuscript. We highly appreciate your time and efforts. The manuscript has been modified according to your comments. The detailed responses are followed in below point by point.

Responses to the comments of the reviewer

Point 1: “Add/Replace the name of the study area to the Keywords.

Response 1:

Thank reviewer for the good suggestion. We have added the name of the study area to the Keywords. (line 43-44, page 1)

“Keywords: SWAT; Ice-melt; Flood processes; Accumulated temperature; NDVI; Tizinafu River Basin (TRB);”

Point 2: “In the last paragraph of the Introduction, the authors should clearly mention the weakness point of former works (identification of the gaps) and describe the novelties of the current investigation to justify us the paper deserves to be published in this journal.

Response 2:

Thanks reviewer for the consideration. We are sorry to describe that inappropriately. We have revised the last paragraph of the introduction as you suggested. (line 91-97, page 2)

“In past, very few studies have been able to perform the hourly scale processes of snow melting and ice melting floods in the alpine region under complex climate conditions. In order to overcome these issues, the topography and climate characteristics of alpine area are fully considered in the discrimination of precipitation form, and the influence of accumulated temperature on precipitation form was added into the new modular.”

Point 3: “In the Tables, highlight values that are more important and discuss them for better understanding readers.

Response 3:

Your good advice was very much appreciated. We have highlighted the important content in the tables and added some descriptions.

Table 1. The LUCC  statistics information table in the study area.

LUCC

Area(km2)

Percentage(%)

Glacier and snowfield

469.96

8.35

Bare soil

815.54

14.49

Bare rock

324.38

5.77

Meadow

1249.17

22.20

Sparse grass

1324.22

23.54

River

69.67

1.24

Marsh

5.65

0.10

Evergreen coniferous shrub

3.98

0.07

Grassland

1202.74

21.38

Broadleaved deciduous forest

3.49

0.06

Evergreen needleleaved forest

138.68

2.46

Dryland

17.42

0.31

Settlement place

1.51

0.03

(line 166-167, page 5)

Table 3. Deviation statistics of peak flood processes before and after model modification.

Date

Deviation of flood peak value(m3·s-1)

Deviation of time(h)

SWAT

SWATAI

SWAT

SWATAI

2013.5

14.52

3.88

7

0

2013.6

63.13

8.83

8

-1

2013.7

14.33

11.59

3

-1

2013.8

57.05

9.7

-1

0

2013.9

-2.88

-2.16

2

1

2014.5

52.7

5.04

-8

2

2014.6

11.23

4.37

-6

0

2014.7

26.25

-2.29

0

0

2014.8

96.05

26.94

-2

1

2014.9

-10.15

-3.03

4

-1

(line 341-342, page 12)

Table 4. Important parameters in the model calibration process.

File extension

Parameter

Description

Range of values

Daily simulation calibrated value

Sub-daily simulation calibrated value

.bsn(New)

ETSI

Elevation threshold between snow and ice

2000–6000

3500

3500

.bsn(New)

IMTP

Ice-melt base temperature

-40

5.39

5.87

.bsn(New)

IMTP_A

Ice-melt base accumulated temperature

0–40

24.68

26.77

.bsn(New)

IMFMX

Maximum melt rate for ice during the year

0–20

12.35

13.04

.bsn(New)

IMFMN

Minimum melt rate for ice during the year

0–20

15.87

16.55

.bsn(New)

ITIMP

Ice temperature lag factor

0–1

0.61

0.65

.bsn(New)

SFTMP_A

Snowfall accumulated temperature

0–40

28

29

.bsn

SFTMP

Snowfall temperature

-20 to 20

3.27

3.46

.bsn

SMTMP

Snow-melt base temperature

-20 to 20

3.06

2.85

.bsn

SMFMX

Maximum melt rate for snow during the year

0–20

7.62

7.57

.bsn

SMFMN

Minimum melt rate for snow during the year

0–20

9.4

8.19

.bsn

TIMP

Snow pack temperature lag factor

0–1

0.55

0.54

.bsn

SNOCOVMX

Minimum snow water content corresponding to 100% snow cover

0–500

38.33

37.84

.bsn

SFTMP

Snowfall temperature

-40

3.36

3.47

.bsn

SURLAG

Surface runoff lag time

0.05–24

11.78

11.43

.gw

ALPHA_BF

Base flow alpha factor (days)

0–1

0.15

0.18

.gw

GW_DELAY

Groundwater delay (days)

0–500

222.68

224.12

.gw

GWQMN

Threshold water depth in the shallow aquifer required for return flow to occur (mm)

0–5000

1175.84

1205.64

.gw

SHALLST

Initial water depth in the shallow aquifer (mm)

0–50,000

4903.68

4958.74

.gw

GW_REVAP

Groundwater “revamp” coefficient

0.02–0.2

0.06

0.05

.mgt

CN2

SCS runoff curve number

35–98

70.79

72.35

.ohru

OV_N

Manning’s “n” value for overland flow

0.01–30

10.77

11.12

.ohru

ESCO

Soil evaporation compensation factor

0–1

0.37

0.35

.ohru

EPCO

Plant uptake compensation factor

0–1

0.39

0.32

.rte

CH_N2

Manning’s “n” value for the main channel

-0.01 to 0.3

0.01

0.01

.rte

CH_K2

Effective hydraulic conductivity in main channel alluvium

-0.01 to 500

47.38

48.12

.sol

SOL_K

Saturated hydraulic conductivity

0–2000

861.31

874.58

.sol

SOL_AWC

Available water capacity of the soil layer

0–1

0.32

0.36

.sub

PLAPS

Precipitation lapse rate

-20 to 20

-5.5

-5.36

.sub

TLAPS

Temperature lapse rate

-10 to 10

-7.59

-7.64

.sub

CH_N1

Manning’s “n” value for the tributary channels

0.01–30

5.42

5.13

.sub

CH_K1

Effective hydraulic conductivity in tributary channel alluvium

0–300

295.67

271.36

.sub

SNO_SUB

Initial snow water content

0–150

95.39

97.33

(line 361-362, page 12-14)

Point 4: “Discuss the reasons of the better performance of the models for drylands compared to the paddy fields.

Response 4:

Thank you very much for your valuable suggestion. We have added relevant content in the discussion section based on your suggestion. (line 584-597, page 21)

“The drylands and paddy fields in the study area were mainly distributed in the lower reaches. In winter, there is lack of water resources. When the temperature rises in spring, the snow and ice melt quickly recharged the river, and the runoff increased continuously. The farmlands mainly depend irrigation water due to the sparse rainfall in the downstream area. Winter wheat and walnut were the main crop types in dryland, and rice was the major crop in paddy field [79,80]. In spring, when winter wheat was irrigated with river water, the wheat will turn green rapidly. Hereby, NDVI value in dryland has a strong sensitivity to the stream flow, or in other words, melting water. The reforestation of walnuts and other crops was also concentrated in April. While, the grow season of rice in paddy field begins in late May with huge water demand. Therefore, NDVI of dryland field vegetation was highly correlated with the amount of melting water. The NDVI and melting water from March to May in spring were used for correlation analysis. During this period, the growth of rice was very slow, and the effect of water storage in paddy field has a certain delay on the response of melting water, which resulted in the correlation between NDVI and melting water in paddy field was slightly lower than that in dry land.[79,80]”

Point 5: “Add R-squared values for comparison between the simulated flood peak values and the measured data of all 25 flood events during the model calibration and validation periods and discuss on the values obtained

Response 5:

Thanks reviewer for the valuable proposal. We have added R-squared values for comparison between the simulated flood peak values and the measured data. (line 501-512, page 19)

“In order to compare the correlation between the simulated and measured flood peaks before and after the model modification, R-squared values were calculated for the 25 flood peaks during the calibration and validation periods. By calculation, the R-squared value between the simulated peaks of the original model and the measured peaks was 0.85 during the calibration period, and that was 0.9 in the modified results. During the verification period, the correlation between the peak value of the original model and the measured data was 0.83, which increased to 0.89 by modifying the model. Through the modification, whether in the calibration period or validation period, the simulated value of the modified model has a higher correlation with the observed data than the simulated values. This was mainly because through the modification of the model, the simulation of the flood peaks were closer to the measured data, which improved the phenomenon that the original model generally underestimated the flood peaks.”

Point 6: “What are the strategies/recommendations to reduce uncertainties in this study?

Response 6:

Thanks reviewer for the consideration. (line 547-557, page 20)

“Uncertainty mainly comes from the input data, model structure, and parameter[55,66]. When reducing the uncertainty of the input data, the model input data was usually compared and screened to remove outliers[67,68]. The most accurate and suitable data was choosedn, and verified to a certain extent. When reducing the uncertainty of the parameters, we usually perform repeated calibration with the help of the calibration tool (SUFI-2) to ensure the optimal parameter value and parameter combination[69-71]. As for the uncertainty of the model structure, it can only be found in practical applications, and the model structure is continuously improved. In this study, through the distinction between snow melting and ice melting calculation, a new ice melting calculation module was added, the structure of the model was optimized, and the simulation results were more accurate. Compared with the original model, the uncertainty of the model structure has been improved.”

 Point 7: “Who can we use the SWATAI model for long term scales (i.e. daily, weekly, monthly, annual)?

Response 7:

Thank you for pointing out this problem. (line 448-455, page 18)

“The SWATAI model can choose different simulation scales according to different research purposes. In the context of future climate change, the annual scale can be used to analyze the relationship between snow, glaciers area and runoff. In the research of flood disaster, hourly scale is a better choice. The daily scale and monthly scale can be used in the research of water resources management. research on water resources management can use daily and monthly scales. The daily scale and weekly scale can be used in the water resources management for reservoirs and agricultural water.”

Point 8: “How can extend the results in other regions with similar/different climates?

Response 8:

Thanks reviewer for the question.

When the model was developed and modified, the applicability of the model has been considered. In this study, a new ice melting module was added to the original model, but the advantages of the original model that required less input data were also retained, mainly using the easily available input data such as temperature and precipitation. Therefore, the model can be used for calculation and simulation as long as the corresponding input data are provided, even if it is applied to the study area with different climatic conditions. Of course, the thresholds of some parameters need to be determined in this process. The determination method is based on comprehensive analysis of meteorological data, remote sensing, statistical data, and measured data in the study area.

Point 9: “The quality of the language needs to improve by a native English speaker for grammatically style and word use.

Response 9:

Thank you for your reminder. We have polished the language of the article through a professional retouching company.

Round 2

Reviewer 1 Report

All comments have already been well addressed.

Reviewer 4 Report

I appreciate the authors for addressing the comments.